# Implications of Resveratrol in Obesity and Insulin Resistance: A State-of-the-Art Review

**DOI:** 10.3390/nu14142870

**Published:** 2022-07-13

**Authors:** Thomas M. Barber, Stefan Kabisch, Harpal S. Randeva, Andreas F. H. Pfeiffer, Martin O. Weickert

**Affiliations:** 1Warwickshire Institute for the Study of Diabetes, Endocrinology and Metabolism, University Hospitals Coventry and Warwickshire, Clifford Bridge Road, Coventry CV2 2DX, UK; t.barber@warwick.ac.uk (T.M.B.); harpal.randeva@uhcw.nhs.uk (H.S.R.); 2Division of Biomedical Sciences, Warwick Medical School, University of Warwick, Coventry CV4 7AL, UK; 3NIHR CRF Human Metabolism Research Unit, University Hospitals Coventry and Warwickshire, Clifford Bridge Road, Coventry CV2 2DX, UK; 4Department of Endocrinology and Metabolic Medicine, Campus Benjamin Franklin, Charité University Medicine, Hindenburgdamm 30, 12203 Berlin, Germany; stefan.kabisch@charite.de (S.K.); andreas.pfeiffer@charite.de (A.F.H.P.); 5Deutsches Zentrum für Diabetesforschung e.V., Geschäftsstelle am Helmholtz-Zentrum München, Ingolstädter Landstraße, 85764 Neuherberg, Germany; 6Centre for Sport, Exercise and Life Sciences, Faculty of Health & Life Sciences, Coventry University, Coventry CV1 2TU, UK

**Keywords:** resveratrol, obesity, insulin resistance

## Abstract

**Background:** Resveratrol is a polyphenol chemical that naturally occurs in many plant-based dietary products, most notably, red wine. Discovered in 1939, widespread interest in the potential health benefits of resveratrol emerged in the 1970s in response to epidemiological data on the cardioprotective effects of wine. **Objective:** To explore the background of resveratrol (including its origins, stability, and metabolism), the metabolic effects of resveratrol and its mechanisms of action, and a potential future role of dietary resveratrol in the lifestyle management of obesity. **Data sources:** We performed a narrative review, based on relevant articles written in English from a Pubmed search, using the following search terms: “resveratrol”, “obesity”, “Diabetes Mellitus”, and “insulin sensitivity”. **Results:** Following its ingestion, resveratrol undergoes extensive metabolism. This includes conjugation (with sulfate and glucuronate) within enterocytes, hydrolyzation and reduction within the gut through the action of the microbiota (with the formation of metabolites such as dihydroresveratrol), and enterohepatic circulation via the bile. Ex vivo studies on adipose tissue reveal that resveratrol inhibits adipogenesis and prevents the accumulation of triglycerides through effects on the expression of Peroxisome Proliferator-activated Receptor γ (PPARγ) and sirtuin 1, respectively. Furthermore, resveratrol induces anti-inflammatory effects, supported by data from animal-based studies. Limited data from human-based studies reveal that resveratrol improves insulin sensitivity and fasting glucose levels in patients with Type 2 Diabetes Mellitus and may improve inflammatory status in human obesity. Although numerous mechanisms may underlie the metabolic benefits of resveratrol, evidence supports a role in its interaction with the gut microbiota and modulation of protein targets, including sirtuins and proteins related to nitric oxide, insulin, and nuclear hormone receptors (such as PPARγ). **Conclusions:** Despite much interest, there remain important unanswered questions regarding its optimal dosage (and how this may differ between and within individuals), and possible benefits within the general population, including the potential for weight-loss and improved metabolic function. Future studies should properly address these important questions before we can advocate the widespread adoption of dietary resveratrol supplementation.

## 1. Introduction

The prevalence of global obesity (now affecting >650 million people) has tripled over the last 50 years [1]. Being overweight, as a forerunner of obesity, now affects >25% of the human population globally (1.9 billion people) [1]. Global obesity, unprecedented in hominin evolution, is a key driver for much 21st Century chronic ill-health and has a major impact on global healthcare expenditure [2]. Moreover, global obesity reflects an important mismatch of the myriad interactions between our ancient genetic endowment and our modern-day post-industrialized cultures and environments. The latter are typified by an abundance of highly processed and highly calorific but fibre-impoverished “convenience” foods [3], co-existing in a “food-desert” of healthy, highly nutritious, and fibre-enriched foods, particularly for people with a lower socio-economic status [4]. The burden of obesity stems from its association with >50 medical conditions [5]. Some of these obesity-related conditions originate from the physical effects of obesity, including biomechanical disorders (commonly, pain in the knee, hip, and back). Others originate from the implications of living with obesity (including its medical sequelae, highly stigmatized status within our society, and impact on psycho-social functioning [6] and work productivity [7]) on mental health and the development of psychiatric conditions [5]. A major subgroup of obesity-related conditions stems from the adverse metabolic sequelae of obesity, that in turn share insulin resistance as a key underlying pathogenic factor.

The association between obesity and insulin resistance (partially mediated through inflammatory effects) [8], and the role of insulin resistance in the pathogenesis of multiple obesity-related conditions is well-described [9,10,11,12]. These conditions include type 2 diabetes mellitus (T2D), hypertension, non-alcoholic fatty liver disease (NAFLD), dyslipidaemia, obstructive sleep apnoea (OSA), and polycystic ovary syndrome (PCOS) [5,13]. Insulin resistance may also underlie pathogenically some obesity-related malignancies [5]. Collectively, obesity-related insulin resistance and its adverse metabolic sequelae underlie much cardiovascular disease [14], which in turn is an important contributor to obesity-related premature mortality [15]. The central role of insulin resistance as a mediator of much of the metabolic burden of obesity, provides a rationale for exploring novel therapeutic strategies for obesity management (including those that target the insulin pathway).

Broadly, the management of obesity has an important focus on dietary change which is usually the first-line approach. Thereafter, other options include bariatric surgery [16] and/or medical therapies [17], much of the latter based on our current understanding of the hypothalamic regulation of appetite and metabolism [18]. Ultimately (and regardless of the approach, including surgical and medical options), effective management of obesity must also include a change of behaviour (often termed “lifestyle management”) [19]. Indeed, many people managed for obesity only receive behavioural change strategies (including dietary and psychological support). The importance of behavioural change to an effective weight-loss strategy cannot be over-stated. Although many aspects of behaviour are important for weight-loss, our diets and eating-related behaviours are particularly relevant. Unfortunately, despite decades of research, our understanding of an ‘optimal’ diet remains incomplete, and there are multiple problems with the execution and interpretation of data from human dietary-based studies [20].

Much of the existing literature on nutrition focuses on macronutrients found in fruit and vegetables, essential vitamins, minerals, and fibre [3,21,22,23,24]. However, in recent years there has been a focus on the potential health benefits of polyphenols (belonging to the class of plant-derived phytochemicals), particularly regarding their effects on non-communicable diseases (NCDs) [21,25]. Polyphenols are distinct from vitamins, minerals, and macronutrients in that their deficiency does not result in any specific deficiency disease, resulting in challenges for recommended dietary intake [21]. One such polyphenol that has received an inordinate amount of attention in recent years (with >10,000 reports in the literature) is resveratrol. Despite such intense interest, there remain many unanswered questions regarding dietary resveratrol, including its activity at the nanomolar range, accumulation within target tissues, biological effects (including its metabolites) within target tissues, modulation of its protein targets, reproducibility of biological effects between individuals, and optimal therapeutic intake [26]. In this review, we discuss the background of resveratrol (including its origins, stability, and metabolism), and its metabolic effects, particularly regarding implications for the management of obesity, T2D and insulin resistance (outlined in Figure 1). We explore the possible mechanisms by which resveratrol may exert its metabolic effects. Finally, we provide concluding remarks and future directions for research into resveratrol, to enable public health recommendations regarding dietary resveratrol supplementation.

## 2. Methodology

We performed a narrative review of the literature using Pubmed for this purpose, and the following search terms: “resveratrol”, “obesity”, “Diabetes Mellitus”, and “insulin sensitivity”, “insulin resistance”. We focused primarily on articles written in English, and on original research based on its importance and relevance to the field. Given the sheer volume of published articles on resveratrol (>10,000 reports in the literature), and the limitation in the number of references permitted for our review, our selection of references was necessarily restricted somewhat. Therefore, we focused on the most impactful and clinically relevant reports and included published review articles where relevant. Although we placed no restrictions on the date of publication, where appropriate, we focused on more recently published articles.

## 3. The Background of Resveratrol

### 3.1. Origins

Polyphenols form a large group of bioactive phytochemicals, including flavonoids and lignans [21]. The stilbenes form one important sub-class of polyphenols. Resveratrol (3,5,4′-trihydroxy-*trans*-stilbene) is a polyphenol (phytoalexin) stilbene molecule (with *cis* and *trans* configurations), that occurs naturally in many plants [27]. Although originally detected in the roots of the white hellebore *Veratrum grandiflorum* in 1939 [27], resveratrol has since been detected in numerous plant species (within roots, stems, flowers, leaves, seeds, and fruits) that include blueberry, grapes, peanut, and cranberry [27,28]. Most notably, the skin of red grapes (and consequently red wine) is the main extract origin of resveratrol [27]. Resveratrol is produced by many plants in response to stressful stimuli that include physical injury or infection from bacteria or fungi [27,28,29]. Although naturally occurring, there have been numerous attempts to chemically synthesise resveratrol through techniques that include the ‘Heck-, Perkin- and Wittig-reactions’ [27]. However, there are potential environmental and cost implications that limit the implementation of such chemical syntheses on a broader scale.

### 3.2. Stability

One of the problems of resveratrol from a dietary perspective is its inherent instability. Unfortunately, resveratrol is sensitive to increased temperature, pH, and light resulting from the instability of its underlying molecular structure (C-C double bond and hydroxyl groups) [27]. Understandably, there has been much interest in improving the stability of resveratrol, to enable its broader usage in functional foods and nutraceutical supplements. One potential solution is the usage of *trans*-resveratrol, which is stable in acidic conditions and at room temperature [27,30]. Other potential solutions to improve the molecular stability of resveratrol include its co-encapsulation with other bioactive molecules such as α-tocopherol, or the formation of resveratrol nanosuspensions [27,31]. Cyclodextrin modification of resveratrol can also enhance its photo-stability [27]. The use of micron-scale grape skin powder was recently reported as a novel approach to further improve the stability and bio-accessibility of trans-resveratrol [32].

In addition to enhancing the stability of resveratrol, other studies have focused on modifying and improving its biological activity through changes to its molecular structure. Although beyond the scope of this review article, briefly, these strategies include modification of: (i) hydroxyl groups (with improvements in antibacterial [33] and cytotoxic activities [34]); (ii) the benzene ring (with improved cytotoxicity in human cancer cell lines [35] and active antioxidant properties [36]), and; (iii) the double bond (with improved anti-cancer and antioxidant properties [37,38]).

### 3.3. Metabolism

Although the level of resveratrol in foods is generally quite low (typically in the lower milligram range in most dietary sources), it is quite variable and influenced by food sources and seasons [39]. Furthermore, although the majority (around 70%) of dietary resveratrol is absorbed following its ingestion, only a tiny minority (0.5%) of this ingested resveratrol becomes bioavailable systemically [40]. Following ingestion, there is rapid and extensive biotransformation of resveratrol within the gastrointestinal tract, and subsequent systemic distribution into organs [26]. It is beyond the scope of this review to provide a detailed discussion of the metabolism of resveratrol, which has been outlined elsewhere [26]. In brief, resveratrol is perceived as a xenobiotic within the gut and crosses into enterocytes that line the small intestine, where it undergoes metabolism (including conjugation with sulfate and glucuronate) with the generation of polar metabolites (to optimize excretion) [26,41]. (A rat study showed that the vast majority of resveratrol [98.5%] is conjugated with only 1.5% entering the bloodstream unmodified [41]). The enzymes responsible for such phase II metabolism (including sulfotransferases and glucuronosyltransferases) manifest genetic polymorphisms, which likely underlies some of the inter-individual variability in the biological effects of resveratrol and its metabolites [26,42]. Following its conjugation within the enterocyte, resveratrol sulfates and glucuronides can either re-enter the intestinal lumen through the apical membrane or enter the bloodstream via the basolateral membrane [26]. Conjugated resveratrol is transported into the blood from the enterocyte through binding to an “Adenosine Triphosphate (ATP)-binding cassette transporter” termed “Multidrug-Resistance Protein 3 (MRP3)” [26,41]. Within the blood, resveratrol and its conjugated metabolites bind to blood proteins (including albumin, haemoglobin, and lipoproteins) to facilitate their vascular transport via the liver, and then systemically to reach peripheral tissues [26].

In addition to phase II metabolism within the enterocyte, polyphenol compounds such as resveratrol and its metabolites also undergo further metabolism within the gut through the action of the gut microbiota, thereby producing bioactive compounds. (One example of such a process is the gut microbiota-induced metabolization of chlorogenic acid [a phenolic compound in the diet] into caffeic acid, with further metabolism producing phenylacetic, phenyl-propionic, and benzoic acid derivatives that are absorbed [21,43]). Regarding the action of the gut microbiota on resveratrol, these processes include hydrolyzation (thereby re-generating resveratrol) or additional reduction reactions [26]. Resveratrol undergoes reduction to Dihydroresveratrol (DHR) through the effects of gut microbiota acting on the double bond between the two phenol rings within the resveratrol molecule [26]. As with the conjugation of resveratrol, the production of reduced metabolites of resveratrol such as DHR (and lunularin) also manifests inter-individual variation [44]. Such resveratrol microbial metabolites are absorbed and either undergo further metabolism within the liver and other tissues, excretion, or enterohepatic circulation via the bile [26]. In a recently reported study using an in vitro epithelial model with Caco-2 cell lines, there were significant differences between the metabolism of trans-resveratrol, cis-resveratrol, and DHR. Whilst trans-resveratrol was transported primarily unchanged through Caco-2 cells, cis-resveratrol was mostly metabolized (via colonic microbiota). Furthermore, sulphate and glucuronide conjugates were the main metabolites of trans-resveratrol and cis-resveratrol/DHR respectively [45].

Having reviewed the background of resveratrol, including its origins as a natural plant source, its inherent instability, potential chemical modifications, and metabolism following its ingestion, it is important to explore the current evidence for the metabolic effects of resveratrol.

## 4. The Metabolic Effects of Resveratrol

Following its discovery 83 years ago, the story of the potential health benefits of resveratrol originates a few decades later, in the 1970s, with data from epidemiological studies that revealed the cardioprotective effects of wine [46]. Since then, evidence has accumulated to support a diverse array of health benefits of resveratrol, including anti-inflammatory and anti-oxidant effects, mimicking calorie restriction, and improved vascular and cellular functioning [26]. Here, we consider the evidence for the metabolic effects of resveratrol in ex vivo adipose tissue cultures, animal- and human-based studies of obesity, and finally, on insulin sensitivity and glycemic control.

### 4.1. Ex Vivo Adipose Tissue Culture

Resveratrol has been demonstrated to have multiple effects on the functioning of human adipocytes ex vivo. These include enhanced lipolytic activity [47], inhibition of adipogenesis (through diminishment of the transcriptional activity and stability of Peroxisome Proliferator-activated Receptor γ [PPARγ]) [48], and prevention of triglyceride accumulation (through enhancement of the expression of sirtuin 1 [SIRT1, a regulator of mitochondrial homeostasis and cellular metabolism]) [49]. In addition to these effects on lipolysis and adipogenesis, there is also evidence that resveratrol has anti-inflammatory effects within adipocytes [50]. This includes resveratrol-induced suppression of a high-fat diet (HFD)-related inflammatory response through downregulation of proinflammatory cytokines (including Interleukin-6 [IL-6], Tumour Necrosis Factor-α [TNF-α], Interferon-α [IFN-α] and Interferon-β [IFN-β]) in a murine model [51], and inhibition of adipose tissue inflammation (and leptin) in human adipose tissue [52,53].

### 4.2. Animal-Based Studies of Obesity

In a rodent-based model, dietary resveratrol diminished HFD-induced weight-gain and obesity through enhanced energy expenditure, mediated in part through the stimulation of fatty acid oxidation within mitochondria, suppression of fatty acid synthesis [51,54], and the induction of brown-like adipocyte formation within white adipose tissue [55]. In a further rodent-based model, resveratrol induced antioxidant effects through the diminishment of diet-induced oxidative stress within white adipose tissue, at least in part through effects on the levels of superoxide dismutase (Sod2) and SIRT1 [54,56]. Finally, in a study on adult rhesus monkeys given a HFD, resveratrol manifested anti-inflammatory effects, with reduced mRNA levels of TNF-α, IL-6, Interleukin-1β (IL-1β) and adiponectin, and suppression of the activation of Nuclear Factor Kappa light chain enhancer of activated B cells (NF-ĸB) within visceral adipose tissue [54,57].

### 4.3. Human-Based Studies of Obesity

The use of resveratrol in human obesity has been the focus of several clinical trials, some of which are still ongoing [54,58]. Despite these studies, none were designed a priori to assess the effects of resveratrol on weight-loss in obesity. In one study in obese men, it was shown that resveratrol treatment (150 mg per day for 30 days) resulted in changes in adipose tissue morphology, including a reduction in the size of abdominal subcutaneous adipocytes [59]. Transcriptome profiling on the adipose tissue showed an enrichment of genes implicated in the regulation of the cell cycle, suggesting enhanced adipogenesis [26,59]. In a separate study on the effects of resveratrol (at the same dose) in obese men, there were further metabolic improvements, including a reduction in the serum levels of inflammatory markers, an increase in energy expenditure, reduced lipolysis and plasma fatty acid, and glycerol levels [60]. However, controversy within the literature regarding the metabolic benefits of resveratrol in obesity remains, particularly when used at higher doses: in one study, resveratrol was used at a dose of 1500 mg per day (an order of magnitude greater than that used in the other two studies outlined here) and failed to show any metabolic effects (including adipose tissue content and energy expenditure) [61].

More recently, the effect of resveratrol on the human adipose tissue metabolome was studied in male subjects with metabolic syndrome, who were treated for four months with resveratrol (1 g) [62]. Using an untargeted metabolomic approach, 282 metabolites were identified within the adipose tissue, of which 45 metabolites changed significantly in response to resveratrol (including increased polyunsaturated and long-chain fatty acids and reduced steroids) [26,62].

### 4.4. Human-Based Studies of Insulin Sensitivity and Glycemic Control

Some of the best evidence for the metabolic benefits of resveratrol in human-based studies relates to its effects on glycemic control and insulin sensitivity in the context of metabolic dysfunction. In a randomized, placebo-controlled crossover study in overweight and obese participants (*n* = 29), it was shown that a co-formulation of trans-resveratrol and hesperetin (tRES-HESP) at concentrations achieved clinically, significantly increased the expression and activity of glyoxalase 1 (Glo1, an enzyme that catalyzes the metabolism of the reactive glycating agent, methylglyoxal) [63]. Furthermore, tRES-HESP significantly reduced the plasma levels of methylglyoxal, total body methylglyoxal-protein glycation and fasting and postprandial plasma glucose, and improved insulin sensitivity, arterial dilatation, and vascular inflammation [63]. Further analysis revealed that tRES-HESP also decreased the activation of the unfolded protein response and expressions of thioredoxin interacting protein (TXNIP) and TNF-α [64]. The authors hypothesized that the co-formulation, tRES-HESP could provide a treatment option to improve the metabolic and vascular health of people who are overweight or obese [63]. In a further randomized, double-blinded placebo-controlled parallel group study in patients with T2D (*n* = 110), the metabolic effects of resveratrol (200 mg/day) versus placebo for 24 weeks were reported [65]. Resveratrol supplementation resulted in significant improvements in glycemic control and insulin sensitivity, improved chronic inflammation and oxidative stress, and associated microRNA expression [65]. In addition to the glycemic pathway, resveratrol may mediate other metabolic effects within the adipose tissue. Recently, it was demonstrated that resveratrol markedly reduces the expression of Angiotensin Converting Enzyme-2 (ACE2) within human adipose tissue. The authors hypothesized that this effect may impact the spread of COVID-19 through the inhibited entry of the SARS-CoV-2 virion into cells (via the ACE2 cell-surface receptor) [66].

Further support for a potential future role of resveratrol supplementation to improve insulin sensitivity and glycemic control stems from a systematic review and meta-analysis of the currently published data, with the inclusion of randomized controlled trials (*n* = 9) that reported on resveratrol in >280 participants with T2D [67]. Resveratrol significantly reduced both fasting plasma glucose and insulin levels (mean reductions of −0.29 mmol/L and −0.64 U/mL, respectively), and significantly improved insulin sensitivity (measured by the homeostasis model assessment of insulin resistance [HOMA-IR]) and blood pressure, with more favourable metabolic effects associated with a higher dose of resveratrol (≥100 mg/day) [67]. However, there were negligible effects of resveratrol on low-density lipoprotein cholesterol (LDL-C), high-density lipoprotein cholesterol (HDL-C), and haemoglobin A1C (HbA1C) [67]. In a further meta-analysis on the effects of resveratrol on glucose control and insulin sensitivity in randomized controlled trials (*n* = 11) and with the inclusion of >380 participants, resveratrol consumption significantly reduced fasting glucose, insulin, HbA1C, and insulin resistance in those with T2D [68]. However, there were no significant effects of resveratrol on glycemic measures in the nondiabetic participants in this meta-analysis [68].

More recently, an umbrella review of meta-analyses of randomized controlled trials provided an overview of the effects of resveratrol supplementation for cardiometabolic risk factor management in patients with T2D, metabolic syndrome, and NAFLD [69]. The inclusion of a total of 38 meta-analyses with >2470 individuals with T2D, metabolic syndrome, and NAFLD revealed favourable and clinically important effects of resveratrol in the short-term on HbA1C [69]. Furthermore, there were beneficial effects of resveratrol supplementation on outcomes such as lipid profile, blood pressure, glycemic control, insulin sensitivity in T2D, waist circumference in metabolic syndrome, and inflammatory markers and body weight in NAFLD. However, for most of these outcome measures, the magnitude of the effect of resveratrol was trivial, with very low to low certainty of evidence [69].

Despite the beneficial metabolic effects of resveratrol outlined in these sub-sections on human-based studies, it is important to counter this with reports of negative effects of resveratrol from the literature, thereby creating some controversy within the field. In one double-blind randomized crossover study on men (*n* = 8) with reduced insulin sensitivity, using ^18^F-fluoroxyglucose Positron Emission Tomography/Computed Tomography (PET/CT) imaging following 34 days of placebo versus resveratrol (150 mg/day) treatment, there was no effect of resveratrol on arterial or systemic inflammation [70]. In a further parallel-group, double-blind study on overweight men and women (*n* = 41), participants were randomized to receive either resveratrol (150 mg/day) or placebo for a period of six months [71]. Although resveratrol was associated with a significant improvement in HbA1C, compared with the placebo group, there was no effect of resveratrol on insulin sensitivity, or on measures of body composition, blood pressure, physical performance, energy metabolism, quality of life, or sleep [71]. Finally, in a large, randomized placebo-controlled trial in overweight and obese insulin resistant human participants for a duration of 12 weeks, it was shown that resveratrol supplementation had no appreciable effects on either liver fat content or cardiometabolic risk parameters [72]. However, the dose of resveratrol (and relatively short duration of supplementation) used in this study may have been too low to have an appreciable effect on liver fat content and cardiometabolic risk parameters [73].

To summarize this section, ex vivo studies on adipose tissue reveal that resveratrol optimizes adipocyte functioning and has anti-inflammatory effects (including through changes in the expression of PPARγ and SIRT1). Indeed, these data represent some of our best evidence to support the metabolically beneficial effects of resveratrol. Although supported by data from animal-based studies, the evidence for metabolic benefits of resveratrol from human-based in vivo studies is relatively depleted [26]. Possible explanations for this apparent disconnect between data from ex vivo and in vivo studies on resveratrol include its pharmacokinetics, with its restricted systemic bioavailability following ingestion as outlined above [40]. Furthermore, there is much inter-individual variation in human responses to oral resveratrol, including its metabolism following absorption and accumulation in tissues (underlined by variations in individual genetic architecture and gut microbiota as described above) [26]. Resveratrol does appear to have some beneficial effects on insulin sensitivity, fasting plasma glucose, and blood pressure in people with T2D, and in the co-formulation of tRES-HESP in those overweight and with obesity. Furthermore, there is some evidence to support the metabolic benefits of resveratrol in human obesity, including adipocyte size, lipolysis, and anti-inflammatory activity. However, there remain important deficiencies in our knowledge and insights, and unanswered questions and controversy regarding the true metabolic benefits of resveratrol for human health, as outlined here:A lack of properly designed studies to assess the weight-loss potential of resveratrol in human obesity and associated metabolic benefits.A lack of studies that properly assess any possible dosage-related effects of resveratrol. Given its complex and variable metabolism, it is possible that the metabolic benefits of resveratrol are dose dependent. Furthermore, metabolically optimal doses of resveratrol may vary between individuals (based on inter-individual differences in resveratrol absorption and metabolism), and within individuals (based on intra-individual differences in environmental factors such as diet).A lack of studies to explore the metabolic benefits of resveratrol in the general population. The current literature focuses primarily on the metabolic benefits of resveratrol in the context of obesity and T2D. Future studies should also focus on the general population (regardless of co-existing conditions like obesity and T2D), including the potential effects of resveratrol on the maintenance of metabolic health and the prevention of weight-gain and metabolic dysfunction.

Having reviewed the current evidence for the metabolic effects of resveratrol across a range of experimental paradigms, it is important to focus on the potential mechanisms by which these effects are mediated.

## 5. The Mechanisms of Action of Resveratrol

Despite the wide body of evidence to support the metabolic benefits of resveratrol, as outlined above, the mechanism(s) of action that underlies the biological effects of resveratrol remain elusive and incompletely understood. Many possible mechanisms have been proposed. These include effects on epigenetic modifications to the DNA sequence (including methylation and histone modifications), thereby influencing gene expression [26,74], and effects of resveratrol as an antioxidant with cellular resistance to oxidative stress (through the induced expression of superoxide dismutase, for example) [75]. In this section, we focus on two possible mechanisms of action of resveratrol that have received the most attention in recent years: interactions with the gut microbiota, and modulation of protein targets.

### 5.1. Interactions with the Gut Microbiota

Polyphenol compounds can modulate the composition of the gut microbiota [76] and appear to have a bi-directional relationship with the human gut microbiota, mimicking their relationship with microbes in the root system of plants [21,77]. Dysbiosis within the gut microbiota associates with many NCDs, including obesity and cardiovascular disease [21,78,79]. It follows, therefore, that polyphenol compounds such as resveratrol may have health benefits through their ability to modify the gut microbiota [21]. The action of resveratrol supplements on the gut microbiota manifests at least in part through acting as an antimicrobial agent [26]. Some polyphenols (including those in black and green tea) inhibit the growth of various detrimental gut microbes (such as *Staphylococcus aureus*, *Pseudomonas aeruginosa*, *Salmonella typhimurium*, *Helicobacter pylori*, and *Escherichia coli*) [21,80]. Resveratrol inhibits the activity of both gram-positive and gram-negative pathogenic bacteria [26] and has known inhibitory effects on bacterial cell growth and cell division in *Escherichia coli* [81].

Polyphenol compounds also stimulate the growth of beneficial gut microbiota (such as *Lactobacillus* spp., *Akkermansia muciniphila*, *Bifidobacterium* spp., and *Faecalibacterium prausnitzii*) [21,76]. Resveratrol has also been shown to modulate the gut microbial composition of obese mice, including an increased relative abundance of *Bacteroides* to *Parabacteroides*, and an improved ratio of *Firmicutes* to *Bacteroidetes* [26,82]. Indeed, by increasing the population of *Bacteroidetes* within the gut of mice, resveratrol supplementation reduces the hepatic production of trimethylamine-*N*-oxide (TMAO), of relevance for a potential mechanism of action of resveratrol given the association of TMAO production with T2D, obesity and cardiovascular disease [26,83]. In a more recently published rodent-based study in the context of an HFD, it was concluded that resveratrol contributed to the prevention of metabolic syndrome through the alteration of gut microbiota and related metabolites and redox status within the intestine [84].

### 5.2. Modulation of Protein Targets

A computational mapping study revealed resveratrol to have the greatest number of all publicly available polyphenol-protein interactions (*n* = 738) [85]. Resveratrol modulates gene expression within pathways of cancer, metabolic and cardiovascular diseases [86]. Indeed, many of the protein superfamilies modulated by resveratrol are relevant for obesity and metabolic diseases, including SIRT1 and proteins related to nitric oxide (NO), insulin, and nuclear hormone receptors (such as PPARγ) [86].

SIRT1 is a caloric restriction mimetic that has been associated with improved longevity and a reduction in age-related complications (such as T2D and obesity) [26]. It is notable that SIRT1 features quite prominently as a protein target of resveratrol. Indeed, a recently reported systematic review of the literature revealed that dietary resveratrol supplementation had beneficial effects on both gene and protein expression of SIRT1 [87]. Amongst its many functions (including the regulation of transcriptional activity, translocation, and histone stability [87]), SIRT1 also plays an important role in the regulation of mitochondrial function, which is likely dependent upon the physiological context and cell type [88]. SIRT1 promotes mitochondrial biogenesis in conditions of energy deficiency and disease/injury but can also promote mitophagy for damaged mitochondria. Although these pathways may occur in parallel, the predominant pathway likely depends on multiple factors that include cellular energy and metabolic status, the presence of SIRT1 substrates, and cellular disease or injury [88]. Prioritization of mitochondrial biogenesis versus mitophagy likely depends on cellular context, with respective examples of terminally differentiated cells such as neurons versus dividing cells [88]. Through the activation of SIRT1, resveratrol deacetylates Peroxisome proliferator-activated receptor Gamma Coactivator 1-α (PGC-1α, a key regulator of cellular metabolism), resulting in the stimulation of mitochondrial biogenesis [26], reduced glycolysis within liver and muscle, and enhanced lipid use [26,89].

Other potential mechanisms of resveratrol action include the inhibition of ATP production through the activation of 5′ Adenosine Monophosphate-activated Protein Kinase (AMPK, a key regulator of energy homeostasis), that in turn inhibits mammalian Target of Rapamycin (mTOR) signaling, with possible anti-ageing effects and improved longevity [26,90]. Resveratrol may improve vascular function and vasoprotective effects, including the stimulation of NO production from endothelial nitric oxide synthase (eNOS) via the actions of AMPK and protein kinase B (Akt) [26,91]. Through inhibition of PPARγ (or via effects on SIRT1), resveratrol enhances lipolysis and reduces adipogenesis [89]. Resveratrol also reduces inflammation through effects on SIRT1 (via repression of NF-ĸB activity) and through interaction with cyclooxygenases (thereby reducing the synthesis of prostaglandins) [26,75]. Finally, through action on the intracellular insulin pathway, resveratrol may have anti-diabetic effects. One study of the flavonoids apigenin and luteolin (related to resveratrol) within a human cell line revealed the triggering of rapid intracellular translocation of the forkhead box transcription factor 01 (FOXO1), which plays an important role in insulin signal transduction [92]. Furthermore, both flavonoids also down-regulated mRNA expression of key enzymes implicated in gluconeogenesis (including glucose-6-phosphatase and phosphoenolpyruvate carboxykinase) and lipogenesis (including acetyl-CoA-carboxylase and fatty-acid synthase) [92]. The effects of these flavonoids on cellular metabolism are schematized in Figure 2. Future studies should explore the effects of resveratrol on intra-cellular cellular metabolic pathways, including the key enzymes implicated in the effects of insulin, and the control of gluconeogenesis and lipogenesis.

## 6. Concluding Remarks

Despite an inordinate amount of scientific interest and published data, including its usage in >140 human clinical trials, the metabolic benefits of resveratrol in humans, including its biological fate and the underlying mechanisms implicated remain elusive [26]. Whilst a systematic review and meta-analysis would provide insights beyond that derived from a narrative review, our choice was to adopt the latter approach given the diversity and sheer breadth of published literature on resveratrol. The bulk of our evidence stems from ex vivo and animal-based studies which show anti-inflammatory and antioxidant effects of resveratrol, including changes in lipid handling within the adipocyte. Although human-based studies also demonstrate some favourable metabolic effects of resveratrol (including anti-inflammatory effects and improved insulin sensitivity in T2D), much of the data discussed in this review lacks novelty, and important unanswered questions remain. Furthermore, we need to remain mindful of the likelihood that our scientific interest in the potential health benefits of resveratrol stem, at least in part, from its notable presence in grapes, and particularly red wine. As an alcoholic drink, red wine is drunk globally to accompany meals, or to mark social events and celebrations, and is often associated with merriment. It is an enticing idea to associate one of life’s pleasures with inherent health benefits, and we need to be cognizant of this as a potential source of bias for any reported study on resveratrol. This is not to dampen in any way our enthusiasm and resolve for the future study of resveratrol, it is merely to remind ourselves of the purpose of such study, and the utmost need for scientific endeavour and methodology. A further concern is the health implications of the alcohol content of red wine. Recent data (with the avoidance of potential biases) reveal that even relatively light alcohol consumption (up to 14 units per week) associates with increased cardiovascular risk within the general population [93]. Furthermore, in a meta-analysis of the data, it was shown that light alcohol consumption increases the risk of malignancy (including cancer of the oesophagus, oral cavity, pharynx, and female breast) [94]. Therefore, any potential benefits and hype from resveratrol consumption within red wine need to be countered and balanced seriously with the detriments to health from the associated alcohol consumption. Given this concern, we should focus on non-alcoholic resveratrol-containing plant-based products as a potential source of dietary resveratrol in future studies.

Of the many unanswered questions regarding the metabolic benefits of resveratrol, perhaps three stand out as being particularly relevant, and worthy of future research efforts. The first relates to the bioavailability of resveratrol and its metabolites. Although free resveratrol has a limited systemic bioavailability following its ingestion, the metabolites of resveratrol (including those metabolites that undergo conjugation within the enterocyte, and reduction within the gut [to form DHR]) likely have similar biological effects, and may act as a pool within the body, and elicit metabolic effects that were previously attributed to free resveratrol [26,95]. If we are to advocate dietary resveratrol supplements within the population, it will be important to acquire full insight into the bioavailability (and biological activities) of both free resveratrol and its metabolites. The second important question relates to the ultimate fates of resveratrol and its metabolites, and the sites of its biological activities. Much of the current literature provides evidence for effects of resveratrol within the adipocyte. However, future studies should focus on other tissues such as muscle and liver to provide a more holistic insight. Achieve this will require novel approaches such as untargeted metabolomics analysis and stable isotopes [26,96] as expedients to elucidate the metabolic fates (and sites of biological activities) of resveratrol and its metabolites. The third important unanswered question relates to the inter- and intra-individual variabilities in the metabolism and biological activities of resveratrol. As outlined, differences between individuals in the gut microbiota and genetic variations (impacting on metabolism pathways) may underlie some of the variability in response to resveratrol between individuals. Other factors such as ethnicity should also be explored [26]. Furthermore, the effects of environmental factors, particularly diet, on the metabolism and ultimate biological activities of resveratrol and its metabolites should be a focus for future research. In the context of obesity, there remain important unanswered questions regarding resveratrol, including its optimal dosage (and how this may differ between and within individuals), and possible benefits within the general population, with potential for weight-loss and improved metabolic function.

Currently, we do not have enough high-quality evidence to advocate widespread and population-wide supplementation with dietary resveratrol. Given the compelling data from ex vivo and animal-based studies, it is important that resveratrol remains a topic for focused human-based research, ultimately to enable evidence-based public health recommendations regarding its ingestion. The metabolic health benefits of moderate red wine consumption (including reduced risk for coronary heart disease) are well described [97]. However, this observation does not prove metabolic causality for resveratrol, given the thousands of other chemicals in red wine including alcohol itself, and other polyphenolic compounds (such as quercetin, catechin, epicatechin and anthocyanin), and numerous other potential confounders. However, perhaps dietary resveratrol supplementation is the wrong question to be asking. After all, our species did not evolve to require resveratrol supplements. If our modern-day, post-industrial diets were not so mismatched to our genetic endowment [98], and we simply reverted to diets that we have evolved to eat, with diverse and plentiful plant-based foods with all its benefits, including those of resveratrol, we would not need to question the need for dietary supplements. Indeed, using resveratrol supplements may simply promote “dysevolution” and compound our poor dietary habits. Therefore, until we have more data, it seems sensible and advisable for us to adopt a diet composed of a diverse array of healthy, unprocessed, and resveratrol-replete plant-based products.

## Figures and Tables

**Figure 1 nutrients-14-02870-f001:**
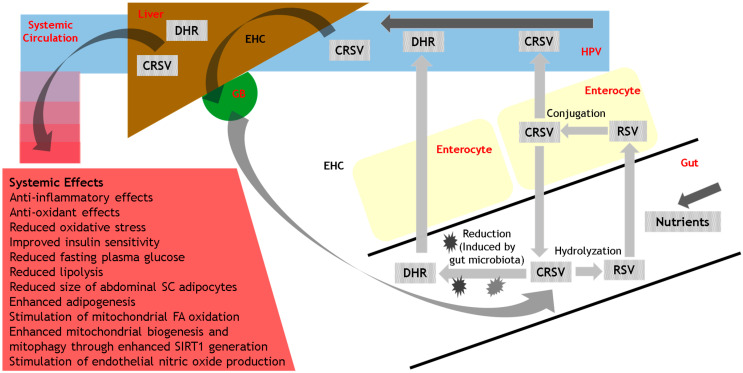
Schematic outline of resveratrol metabolism and biological actions. CRSV = Conjugated resveratrol; DHR = Dihydroresveratrol; EHC = Enterohepatic circulation; FA = Fatty acid; GB = Gallbladder; HPV = Hepatic Portal Vein; RSV = Resveratrol; SC = subcutaneous; SIRT1 = sirtuin 1.

**Figure 2 nutrients-14-02870-f002:**
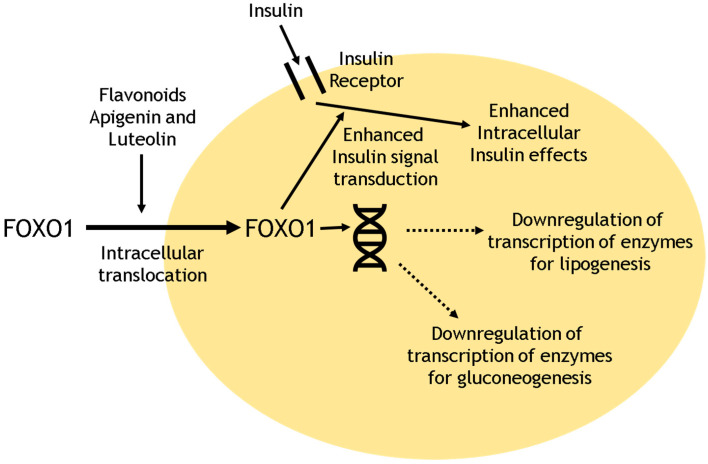
Schematic outline of flavonoid effects on cellular metabolism. FOXO1: Forkhead box transcription factor 01.

## Data Availability

Not applicable.

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
