# Peer review of "Implications of Resveratrol in Obesity and Insulin Resistance: A State-of-the-Art Review"

_nutrients, 2022, doi:10.3390/nu14142870_

Round 1

Reviewer 1 Report

Manuscript presented by Barber et al. provides a better understanding on resveratrol and its metabolic effects on obesity, type II Diabetes and insulin resistance. The concept is interesting, nevertheless, there are some areas that need to be addressed to improve the manuscript. Please, find below several comments:

Some questions and concerns need to be addressed before this manuscript may be considered for publication.

ABSTRACT

LINE 18-19

Red wine is abundant in resveratrol. However, as an alcoholic drink, the alcohol in this product has been associated with several health problems. Since 2013 there is strong evidence showing that light alcohol dinking is a risk factor linked to several diseases and it has been associated with cancer (DOI: 10.1016/j.clnu.2021.12.009 ; DOI: 10.1093/annonc/mds337). I believe, this concept should be clarified.

An overview of existing current research requires some improvement. Considering this is a review article, there are very few current papers cited in the references overall (between years 2020-2022). Please, improve this aspect.

I am not sure of the novelty of the data present, the authors themselves should have mentioned this in the discussion. In my opinion, authors should add a discussion section for this purpose.

Please, reconsider and explain this aspect.

LINE 110-120, Methodology

Methods should be better and widely described

The search has been done in MEDLINE, Embase database or Cochrane Library? And have the authors used any tool or method on the article screening?

Please, reconsider and explain this aspect.

CONCLUDING REMARKS

LINE 441-453

As mentioned above, red wine is abundant in resveratrol, and it also has this social aspect. However, as an alcoholic drink, the alcohol in this product has been associated with several health problems. Since 2013 there is strong evidence showing that light alcohol dinking is a risk factor linked to several diseases and it has been associated with cancer (colorectal, breast, pharynx, larynx, oral, esophagus, liver etc.) (DOI: 10.1016/j.clnu.2021.12.009 ; DOI: 10.1093/annonc/mds337). I believe, this concept should be clarified.

Please, reconsider this aspect.

LINE 447-450

There is no possible comparation between broccoli (and other brassica vegetables) and red wine. Broccoli is not harmful to health, it can not cause any health issue. Red wine has alcohol, which according to wide scientific evidence, is toxic to health.

Please, reconsider this aspect.

The review seems a bit like a missed opportunity. Conducting a Systematic review or a Meta-analysis would add value to the work. Also, it would improve the quality overall. 

Author Response

Reviewer 1

Comment 1.1:

Manuscript presented by Barber et al. provides a better understanding on resveratrol and its metabolic effects on obesity, type II Diabetes and insulin resistance. The concept is interesting, nevertheless, there are some areas that need to be addressed to improve the manuscript. Please, find below several comments. Some questions and concerns need to be addressed before this manuscript may be considered for publication. ABSTRACT, LINE 18-19: Red wine is abundant in resveratrol. However, as an alcoholic drink, the alcohol in this product has been associated with several health problems. Since 2013 there is strong evidence showing that light alcohol dinking is a risk factor linked to several diseases and it has been associated with cancer (DOI: 10.1016/j.clnu.2021.12.009; DOI: 10.1093/annonc/mds337). I believe, this concept should be clarified.

Response to comment 1.1:

Thank you for this comment. We agree with the reviewer that this is a very important point. In response, we have added some additional text with these two new suggested references added, to the concluding remarks section of the revised version of the manuscript. We have also removed the text relating to red wine consumption at the end of the concluding remarks section. Please kindly also see our response to comment 1.6.

Comment 1.2:

An overview of existing current research requires some improvement. Considering this is a review article, there are very few current papers cited in the references overall (between years 2020-2022). Please, improve this aspect.

Response to comment 1.2:

Thank you for this comment. We agree with the reviewer. In response, we have added additional information with new references for studies published within the last 2-years to the revised version of our manuscript. This includes additional text and references to sub-section 3.2 on stability of resveratrol, sub-section 3.3 on metabolism of resveratrol, sub-section 4.4 on human-based studies of insulin sensitivity and glycemic control, sub-section 5.1 on interactions with the gut microbiota, and sub-section 5.2 on modulation of protein targets.

Comment 1.3:

I am not sure of the novelty of the data present, the authors themselves should have mentioned this in the discussion. In my opinion, authors should add a discussion section for this purpose. Please, reconsider and explain this aspect.

Response to comment 1.3:

We agree with the reviewer. In response, we have added some additional text to the concluding remarks section (in which we also include discussion of the data outlined [please see responses to comments 1.1 and 1.6]) of the revised version of our manuscript.

Comment 1.4:

LINE 110-120, Methodology: Methods should be better and widely described.

Response to comment 1.4:

As this is a narrative review on a very broad field, we avoided being too prescriptive in our approach. We provided an overview of how the narrative review was conducted, including search terms and the used search platforms. Please kindly see our response to comment 1.5.

Comment 1.5:

The search has been done in MEDLINE, Embase database or Cochrane Library? And have the authors used any tool or method on the article screening? Please, reconsider and explain this aspect.

Response to comment 1.5:

Thank you for this comment. We refer the reviewer to the methodology section of our original manuscript. As outlined, we used Pubmed as a search platform for our narrative review, using the search terms as stated in our manuscript. As outlined, we focused primarily on articles written in English and chose original research based on its perceived importance and relevance to the field (including those papers published more recently). Given the volume of published research on resveratrol (>10,000 reports in the literature), our choice of references was necessarily restricted. We did not use any formal tool or method for screening of articles, as this was not intended for our purposes.

 Comment 1.6:

CONCLUDING REMARKS. LINE 441-453: As mentioned above, red wine is abundant in resveratrol, and it also has this social aspect. However, as an alcoholic drink, the alcohol in this product has been associated with several health problems. Since 2013 there is strong evidence showing that light alcohol dinking is a risk factor linked to several diseases and it has been associated with cancer (colorectal, breast, pharynx, larynx, oral, esophagus, liver etc.) (DOI: 10.1016/j.clnu.2021.12.009; DOI: 10.1093/annonc/mds337). I believe, this concept should be clarified. Please, reconsider this aspect.

Response to comment 1.6:

We agree with the reviewer that this is a very important point. In response, we have added some additional text with these two new suggested references to the concluding remarks section of the revised version of the manuscript. We have also removed the text relating to red wine consumption at the end of the concluding remarks section. Please also see our response to comment 1.1.

Comment 1.7:

LINE 447-450: There is no possible comparation between broccoli (and other brassica vegetables) and red wine. Broccoli is not harmful to health, it can not cause any health issue. Red wine has alcohol, which according to wide scientific evidence, is toxic to health. Please, reconsider this aspect.

Response to comment 1.7:

We are grateful for the comment. To avoid possible confusion regarding comparisons between brassica vegetables and red wine, we have deleted the text relating to broccoli and brassica vegetables in the concluding remarks section of the revised version of our manuscript.

Comment 1.8:

The review seems a bit like a missed opportunity. Conducting a Systematic review or a Meta-analysis would add value to the work. Also, it would improve the quality overall. 

Response to comment 1.8:

Please kindly see our response to comment 2.5. We agree with the reviewer regarding the general utility of systematic reviews and meta-analyses of the literature, and that this approach can offer insights beyond a mere narrative review of the literature. However, we are not confident that the existing literature on resveratrol (with its complexities, diverse research techniques and settings and shear breadth of focus) lends itself well to such an approach. Performing a systematic review would necessitate the selection of a specific question amongst the many unanswered questions within the field, and would necessarily restrict our approach,. Such an approach would simply not enable the breadth of coverage that a narrative review does. For this reason, we had accepted the invitation from the Editor to provide a narrative review of the general literature on resveratrol, to facilitate the illustration and summary of a very complex and diverse field of published studies. We have added some additional text to the concluding remarks section of the revised version of our manuscript to explain our rationale for choosing a narrative review approach.

Reviewer 2 Report

Dear Editor,

I carefully read the manuscript by Barber et al.

My comments and suggestions are the following:

 - The manuscript (authors' name and references) should be formatted following the instructions for the authors.

 - The authors should specify in the title of the article that this is a state-of-the-art review.

 - The manuscript is well written in standard English. However, the authors should declared all the used abbreviations at their first occurrence. 

 - The authors should consider to refer to doi: 10.1111/dom.13324 in their manuscript.

 - Among the "Conclusion remarks", the authors should also discuss the limitations of their not-systematic approach.

Author Response

Reviewer 2

Comment 2.1:

I carefully read the manuscript by Barber et al. My comments and suggestions are the following: The manuscript (authors' name and references) should be formatted following the instructions for the authors.

Response to comment 2.1:

Thank you for this comment. In the revised version, we have ensured that the author names and references are formatted correctly.

Comment 2.2:

The authors should specify in the title of the article that this is a state-of-the-art review.

Response to comment 2.2:

We have adjusted the title as suggested.

Comment 2.3:

The manuscript is well written in standard English. However, the authors should declare all the used abbreviations at their first occurrence. 

Response to comment 2.3:

Thank you for this comment. In response, we have ensured that in the revised manuscript, all abbreviations are written in full at their first occurrence. 

Comment 2.4:

The authors should consider to refer to doi: 10.1111/dom.13324 in their manuscript.

Response to comment 2.4:

Many thanks. We have added the suggested reference along with additional text to the section on ‘the metabolic effects of resveratrol’ in the revised version of our manuscript.

Comment 2.5:

Among the "Conclusion remarks", the authors should also discuss the limitations of their not-systematic approach.

Response to comment 2.5:

Thank you for this comment. In response, we have added some additional text to the concluding remarks section in the revised version of our manuscript.